# Barriers to Access and Utilization of Diabetes Care Among Patients with Severe Mental Illness in Saudi Arabia: A Qualitative Interpretive Study

**DOI:** 10.3390/healthcare13050543

**Published:** 2025-03-03

**Authors:** Mashael A. Hobani, Lina H. Khusheim, Bedor A. Fadel, Shaima Dammas, Waleed M. Kattan, Mohammed S. Alyousef

**Affiliations:** Department of Health Services Administration and Hospitals, Faculty of Economic and Administration, King Abdulaziz University, Jeddah 80201, Saudi Arabia; mahobani@kau.edu.sa (M.A.H.); lhkhusheim@kau.edu.sa (L.H.K.); bodorfadel@gmail.com (B.A.F.); sh_dammas@hotmail.com (S.D.); wmkattan@kau.edu.sa (W.M.K.)

**Keywords:** diabetes care, mental illness, type 2 diabetes, qualitative study, Saudi Arabia

## Abstract

**Background/Objectives**: The prevalence of Diabetes Mellitus (DM) is a major concern in Saudi Arabia, making it a challenge for health delivery for those with severe mental illness (SMI). This study aims to explore the barriers to access and utilization of diabetes care among patients with diabetes and serious mental illnesses, their relatives, and healthcare providers to provide evidence-based recommendations for health policy improvement. **Methods**: A qualitative interpretive research design was used via Braun and Clarke’s framework to analyze the data thematically. Semi-structured interviews were conducted with 35 participants, including patients, relatives, and healthcare providers between September and October 2023, in Jeddah city, Saudi Arabia. **Results**: The following four themes emerged from the qualitative data: (1) The status of integrated care, (2) Barriers to access to diabetes care at different levels, (3) Navigating obstacles to providing comprehensive diabetes care, and (4) Evidence-based recommendations for health policy improvement. **Conclusions**: This study underscores the necessity for a comprehensive and integrated approach to care, educational programs, specialized clinics, and improved healthcare logistics. Integrating mental health and diabetes management is needed to ensure better utilization.

## 1. Introduction

Diabetes Mellitus (DM) is a burgeoning global health emergency, with an estimated 422 million people affected worldwide, the majority being in lower-income nations [1]. The World Health Organization attributes more than 1.5 million deaths annually to diabetes, a figure that highlights the pressing need for effective management and intervention strategies [1]. Globally, it was estimated that over 1.31 billion of the population will have diabetes by 2050 [2].

In Saudi Arabia, the impact of diabetes is particularly acute, with the condition affecting over 4 million individuals, or 17.7% of the adult population [3]. Jarrar et al. have shed light on the epidemiological landscape of type 2 Diabetes Mellitus (T2DM) in Saudi Arabia [4], noting a pooled prevalence of 16.4% from 2000 to 2020. Their findings indicate not only the pervasiveness of diabetes in the region, but also the significant variability in diabetes prevalence across different age groups and localities, which calls for a targeted and localized approach to diabetes prevention and care. AlJehani et al. [5] found that the prevenance of diabetes was higher in Baha region compared to Jizan region (18.4% vs. 9.5%).

The confluence of diabetes with severe mental illnesses (SMI) amplifies the complexity of care and management, making it an area of significant concern for health systems [3,6]. The intersection of these two conditions is especially critical given that individuals with severe mental illnesses are disproportionately affected by diabetes, a fact that complicates their healthcare needs and outcomes [6]. The National Institute of Mental Health defined severe mental illnesses as “a mental, behavioral, or emotional disorder resulting in serious functional impairment, which substantially interferes with or limits one or more major life activities” [7].

Diabetes management becomes a challenge among those with severe mental illnesses in terms of delivering an effective health care service. Patients with severe mental illnesses are at a higher risk of developing comorbidities, which can complicate diabetes management. The common prevalent comorbidities are cardiovascular disease, asthma, and chronic obstructive pulmonary disease, all of which are known to impact diabetes care [8,9]. Therefore, diabetes management has become a challenge for those with mental illness [8,10]. The impact of severe mental illnesses on diabetes management is profound. For instance, severe mental illnesses can heavily affect treatment adherence and overall care for patients, leading to suboptimal management of diabetes [6]. It was suggested severe mental illnesses can drastically reduce the effectiveness of diabetes care, as they affect both the physical and psychological capacity to manage the condition [11].

Additionally, there are many factors that prevent individuals with severe mental illnesses from effectively managing their diabetes in Saudi Arabia. The existing diabetes self-management and education programs are insufficient due to low health literacy rates and psychological barriers, such as depression, which hinder effective self-management, as well as the lack of structured implementation of diabetes education programs [3]. Moreover, challenges in adopting mobile health solutions and a shortage of mental health support further complicate the issue [7].

The nuances of diabetes management in the context of severe mental illnesses were further explored by Bellass et al. [12] and Dorey et al. [13], emphasizing the need for healthcare providers to recognize and adapt to their patients’ cognitive and emotional states. This understanding can play a pivotal role in enhancing medication adherence and the overall efficacy of diabetes care [14]. The socioeconomic status of patients also emerges as a determinant of health outcomes, with Kremers et al.’s [15] study suggesting that those from lower socioeconomic backgrounds may experience more significant challenges in managing both diabetes and severe mental illnesses.

Although there has been an increase in understanding of the bi-directional association between mental illness and diabetes, studies have shown that patients with mental illnesses have increased levels of diabetes incidence due to inadequate access to healthcare services [10], stigmatization [8], and inadequate training for healthcare providers [6]. Mental health patients with diabetes have a higher likelihood of increased morbidity and mortality due to complication-related deaths associated with diabetes than patients without mental health disorders and cardiac diseases [4]. These issues call for the need to determine the subjective experiences and attitudes of patients, the relatives involved, and healthcare providers.

This research seeks to explore the barriers and facilitators that impact the access to and utilization of diabetes care among individuals with severe mental illnesses in Jeddah province, Saudi Arabia. This study aims to paint a comprehensive picture of the multifaceted obstacles and opportunities within the existing healthcare framework by collating the perspectives of patients, their relatives, and healthcare providers. The objectives are as follows:

To shed light on the barriers that impede effective diabetes management for individuals with severe mental illnesses;To identify and explore facilitators that navigate barriers to access to diabetes care and improve care;To propose actionable recommendations to enhance the quality and accessibility of diabetes care for this vulnerable subgroup.

The importance of this research lies in its potential to fill a critical void in the literature by focusing on the intersection between diabetes care and severe mental illnesses from an integrated, multi-stakeholder perspective. The unique challenges faced by individuals with severe mental illnesses in accessing diabetes care are often underrepresented in healthcare research and practice. By elucidating these challenges, this study contributes valuable insights that can inform the development of integrated care models that are attuned to the complex needs of this population. The findings are poised to have a substantial impact on policy and practice, promoting a healthcare system that is more inclusive, effective, and empathetic, recognizing the intricate interplay between physical and mental health as outlined by Bellass et al. [8] and Dorey et al. [9].

## 2. Materials and Methods

### 2.1. Study Design

This study used a qualitative interpretive approach to explore the participants’ perspectives on the barriers to access and utilization of diabetes care. Semi-structured interviews were conducted to collect data from patients with concurrent diagnoses of severe mental illnesses and diabetes, as well as the relatives involved in patient care and healthcare providers (see Appendix A). These data were analyzed by applying Braun and Clarke’s thematic analysis [15].

### 2.2. Participant Selection and Criteria

In this study, three participant categories were engaged, including patients with diabetes and severe mental illnesses, their relatives, and healthcare providers. The three groups of participants were generally identified from hospitals’ records and databases, patients’ referrals, and physicians’ recommendations. Eligible patients were adults aged 18 years or older, those diagnosed with type 2 diabetes, and those concurrently managing severe mental illnesses. The inclusion criteria necessitated chronic history of severe mental illness, use of medications, mental ability to answer the questions, physical health issues and functional interference in daily life activities and fluency in Arabic or English. The patients were identified from the hospitals’ records.

Relatives were identified via patient referrals, physician recommendations, and hospital records. Eligibility required the relatives to be in regular contact with the patient, knowledgeable about the patient’s health status, and actively involved in monitoring the patient’s diabetes management plan, including providing psychological and financial support to assist the patients with the expenses of living and treatment.

Healthcare provider participants, including physicians and nurses, were sourced from hospital databases. These individuals were directly involved in the treatment, care, and support of patients with both severe mental illnesses and diabetes, operating in both inpatient and outpatient settings.

The sampling was conducted purposively, with the goal of reaching theoretical saturation. Data collection continued until no new themes emerged from the interviews, in line with the principles outlined by DiStefano and Yang [16]. A total of 35 interviews were conducted in Arabic language, since all of the participants spoke Arabic language only, providing a robust spectrum of experiences and perspectives on accessing and utilizing diabetes care among the participant groups. The participants were distributed as follows: 8 patients, 10 relatives, and 17 healthcare providers. Each interview lasted between 40 min and 1 h.

Although there were anticipated concerns regarding achieving saturation, efforts were made to ensure that diversity was realized within each group. First recruitment was carried out on patients, and the interview undertaken identified key foundational themes. These themes were used to guide the development of the relatives’ and healthcare providers’ interviews. As such, this improved the realization of saturation despite the constraints on the number of participants.

### 2.3. Ethical Considerations and Informed Consent

Prospective participants were presented with detailed information about this study’s scope, nature, and potential implications, ensuring informed consent was obtained prior to participation. They were assured of the right to withdraw at any time without penalty, with interviews scheduled at their convenience, either in a dedicated hospital space or via telephone.

Even though the participants with severe mental illnesses might have limited decision-making capacity, their mental capacity to understand the study’s nature, risks, benefits was assessed. However, if they were unable to provide consent, a legally authorized representative or surrogate decision maker gave consent on their behalf. Additionally, the participant’s assent was sought if possible, and their well-being was continuously monitored throughout this study.

Additionally, this study adhered to ethical standards, receiving approval from the Institutional Review Board at the Research and Studies Affairs Unit, Ministry of Health, Saudi Arabia (A01709), underscoring the commitment to ethical research practices.

### 2.4. Data Collection Procedures

The data triangulation method was used to develop a comprehensive understanding of phenomena and to enhance validity and credibility of the findings. In this study, data were gathered through structured, in-depth, face-to-face interviews with three different groups, which were patients, their relatives, and healthcare professionals, in September and October 2023. These interactions occurred in secondary care centers in Jeddah province, Saudi Arabia. A qualitative semi-structured interview was utilized to collect the data from the participants. The interviews were focused on the participants’ experiences and perspectives regarding access to and utilization of diabetes care for individuals with severe mental illnesses. The interviews were started by discussing managing diabetes alongside severe mental illness and then discussing the experiences of healthcare received for diabetes and severe mental illnesses.

Interviews were audio-recorded with the participants’ consent, ensuring accuracy and integrity in data capture. Subsequently, the recordings were meticulously transcribed verbatim. To maintain privacy, all personal identifiers were redacted or altered during the transcription of the collected data to ensure the participants’ confidentiality.

### 2.5. Data Analysis

The collected interview data underwent a rigorous thematic analysis, adhering to the methodological framework established by Braun and Clarke [17]. This analytical process involved several key steps, including familiarizing with the data, generating initial codes, searching for themes, reviewing and refining those themes, defining and naming them, and finally writing up the findings.

The primary analysis was conducted and reviewed in Arabic to maintain the integrity of the participants’ expressions and nuances. Following the thematic identification, relevant data excerpts were translated into English, ensuring that the translation preserved the original meaning and context.

The analytical phase was iterative, with the research team continually reviewing and refining the themes to ensure a robust and representative data synthesis. This approach facilitated a nuanced understanding of the complex experiences and challenges encountered by the participants in accessing and utilizing diabetes care within the context of severe mental illnesses. The researcher employed member checking, which was conducted to ensure the trustworthiness of the findings. These findings were shared with the research group to give their perspectives and ensure the accuracy of the results.

## 3. Results

### 3.1. Participant Demographics and Characteristics

The qualitative methodology of this study entailed semi-structured interviews as the primary instrument for data collection. These interviews, which were employed to elicit in-depth insights into participants’ experiences, included a combination of demographic queries and key questions tailored to each distinct segment of the sample.

The sample comprised 35 participants, including the following: 8 patients with concurrent diagnoses of severe mental illnesses and diabetes, 17 healthcare providers, and 10 relatives involved in patient care. The patient cohort included an equal gender distribution of four men and four women, with the predominant diagnosis being schizophrenia (n = 6), followed by bipolar disorder (n = 2). All patient participants were diagnosed with type 2 diabetes and ranged in age from 26 to 81 years. The employment status was diverse, with half being unemployed and three being retired (Table 1).

The relative participants were primarily siblings (n = 8), with the remainder being a daughter and a cousin. Emotional and financial support was common among these relatives, as can be seen in Table 2.

Seventeen healthcare providers, encompassing three physicians and fourteen nurses, were included in this study. This group demonstrated significant professional experience, averaging ten years in their respective roles, and included mostly female participants (n = 12). These details are presented in Table 3.

### 3.2. General Charactrstics of the Participants

The thematic analysis of the interview transcripts followed a systematic approach, commencing with a thorough familiarization with the data. This was followed by a meticulous transcription process and rigorous coding to identify emergent themes. Four central themes were derived from the data, reflecting the lived experiences of patients who have diabetes with severe mental illnesses and the perspectives of their healthcare providers and relatives.

The participants were from both inpatient and outpatient clinics from secondary care centers in Jeddah province, Saudi Arabia, contributing approximately eight hours of dialogue spanning both morning and evening sessions. This diverse temporal distribution ensured a broad capture of daily variations in participant experiences and interactions within the healthcare environment.

The participating patients, with an average age of 46 years, were predominantly educated up to high school level and had been living with a diabetes diagnosis for approximately 25 years on average. Insulin dependency was a commonality among all 35 individuals with diabetes in this study since diagnosis, with all individuals receiving care from immediate or extended family members.

The relatives of these patients, integral to the treatment continuum outside of formal healthcare settings, were interviewed to delineate their challenges. This perspective was crucial in understanding the multifaceted nature of caregiving and support dynamics.

The healthcare providers, with a substantial average tenure in the medical field, highlighted five primary roles in delivering care to this patient demographic, as follows: patient reception, medication prescription, education and awareness provision, follow-up, and equipping patients with the necessary skills and knowledge for self-management.

### 3.3. Thematic Analysis and Emerging Themes

Four central themes were derived from the data, as follows: (1) The status of integrated care, (2) Barriers to access to diabetes care at different levels, (3) Obstacles to providing comprehensive diabetes care, and (4) Evidence-based recommendations for health policy improvement.

#### 3.3.1. The Status of Integrated Care

The status of an integrated care approach theme encapsulates the systematic care provided to people who have diabetes with severe mental illnesses from the moment they enter the healthcare facility. Medical team members detailed a thorough patient intake process, which involved conducting initial examinations, necessary investigations such as reviewing medical histories, and engaging with patients and their families or caregivers to understand their current health condition.

The participating doctors agreed on their pivotal role in determining the appropriate medications after assessing the patient’s condition and considering any previous treatments. One of the central aspects of patient care identified was the provision of educational information about the health condition and treatment plan. This education is crucial not only for patients, but also for their relatives, as follows:

“*Directly instructing the patient can be challenging, so we focus on educating the families and relatives. We advise them to treat the patient with the care one would offer a child*” (HF2)

Physicians underscore the critical nature of consistent follow-up appointments for the sake of meticulously reviewing and adjusting patients’ medication regimens as needed. Nurses also play a vital role in ensuring that patients maintain their treatment regimen and diet. They monitor patients, provide emotional support, and promote home care services through their home visits. The relatives acknowledged the positive impact of such services, as follows:

“*My brother’s condition was gravely worrisome, but thankfully, the home care team’s visits significantly improved his situation, offering our family much-needed respite*” (RM1)

Nurses also prepare patients for doctors’ examinations and ensure all necessary medical supplies are ready for use. All healthcare professionals confirmed the provision of periodic training courses, lectures, and workshops for nurses’ education, although the demanding nature of their work sometimes hindered attendance, as follows:

“*With four departments to manage, we doctors find it challenging to leave for lectures. Hence, we take turns attending based on who is available*” (HM4)

One nurse pointed out the importance of training for fulfilling annual professional development requirements. Communication within the medical team is direct, ensuring efficient internal coordination, while external communication for referrals is conducted through the Ministry of Health’s referral system, especially in urgent cases, as follows:

“*In emergencies, we promptly get in touch with the Referral Center of the Ministry of Health through urgent telephone contact*” (HM1)

Overall, the theme presents a multifaceted view of the healthcare providers’ responsibilities, from the clinical management of patients to the critical educational support extended to families, ensuring comprehensive care for patients who have diabetes with severe mental illnesses.

#### 3.3.2. Barriers to Access to Diabetes Care at Different Levels

This theme provides insights into barriers that face the diabetes patient with SMI at different levels. This theme was categories into three main sub-themes, as follows: individual-level barriers, systemic barriers, and social-cultural barriers.

##### Individual-Level Barriers

The resistance of patients who have diabetes with severe mental illnesses to acknowledge their diabetic condition and adhere to medication regimens introduces additional hurdles, with fluctuations in blood glucose levels exacerbating their mental health issues and potentially leading to violent behavior.

This narrative encapsulates the grave implications that diabetes has on the lives of individuals with severe mental illnesses, affecting their mobility and overall quality of life. It also highlights the need for integrated care approaches that address both physical and mental health challenges.

The direct and indirect effects of psychiatric medications underscore the complex interplay between diabetes and severe mental illnesses. Healthcare providers unanimously agree that these medications can significantly increase the risk of developing diabetes, mainly due to their side effects, which include an increased appetite, particularly for sugary foods, and a tendency to induce lethargy. Medications such as Olanzapine and Zyprexa are particularly noted for their substantial impact on patients’ appetites, contributing to weight gain and metabolic disturbances.

“*A particularly severe consequence observed is termed ’Metabolic Syndrome’, characterized by central obesity, which often progresses to diabetes, hypertension, and other related conditions. Remarkably, patients demonstrate significant weight gain over time; for instance, a patient’s weight escalated from 90 kg to 120 kg within a year, underscoring the profound impact of their treatment regimen*” (HF6)

This cascade of effects is further complicated by the inherent challenges faced by individuals with severe mental illnesses, such as an inability to control their behavior, leading to severe physical consequences, including wounds that heighten the risk of diabetes.

“*Numerous patients present in the emergency room bearing extensive wounds, particularly on their feet, attributed to walking barefoot. These injuries, coupled with elevated blood glucose levels, constitute a substantial healthcare challenge*” (HF3)

##### Systemic Barriers

One of the nurses indicated that patients with severe mental illnesses face the challenge of navigating through the fragmented healthcare system, and they are sometimes faced with the financial burden of balancing the mental health treatment with diabetic care.

One of the relatives mentioned that the transportation costs and delays in the hospitals have been a major impediment to access to healthcare services, as follows:

“*Sometimes I feel frustrated due to the delays in accessing services, and I cannot attend to my chores on time. We have to go through all process every time we are here*” (RM1)

##### Social-Cultural Barriers

One member of the medical team indicated the increased stigmatization within the community, as the patients with severe mental illnesses are perceived as personal failings. Such judgment deters them from accessing quality diabetes care. Also, there has been limited community support, and they often find themselves isolated with only a few friends and relatives willing to engage with them.

#### 3.3.3. Navigating Obstacles to Providing Comprehensive Diabetes Care

The medical team delineates the barriers to effective diabetes care within severe mental illnesses contexts into internal and external categories. Internally, a critical shortfall in resources—ranging from advanced diagnostics to essential diabetes medications—elicits a sense of frustration among healthcare providers.

“*Our capabilities in medication are constrained, given our focus on mental health. Psychiatric medications are plentiful, yet we face a stark scarcity of diabetes treatments*” (HM3)

Additionally, the absence of specialized clinics within mental health facilities exacerbates these challenges, necessitating referrals that delay critical care.

“*Referrals for minor treatments, such as a single stitch, underscore our dire need for comprehensive in-house services*” (HM2)

Staff shortages, particularly in nursing, further strain the system, compromising patient care and placing undue burdens on existing staff.

“*With a patient-to-nurse ratio grossly imbalanced, the psychological and physical toll on our nursing staff is profound*” (HM5)

A universal concern is the lack of targeted educational initiatives for patients and their families, which is critical for managing this dual diagnosis effectively.

“*The gap in dedicated educational support significantly undermines our efforts to empower patients and their families in managing diabetes*” (HM3)

Externally, deficits in familial support and behavioral challenges for the patients were reported as obstructions to care continuity and adherence to treatment protocols.

“*Distractions and delusions among our patients with psychological illnesses severely impact their diabetes management*” (HF9)

Despite these hurdles, patients and their families show improvement in care and education. However, misconceptions persist regarding the origins of diabetes and its management.

“*Attributions of diabetes to diet or divine will, without recognizing the role of psychiatric medications, reflect a critical awareness gap*” (PM2)

“*It is from juices and sweet things*” (PF3)

In addition to the misconception regarding diabetes management, there is a lack of knowledge among families regarding the impact of psychiatric treatments on eating habits. For instance, families often remain uninformed about the interconnectedness of diabetes and severe mental illnesses, attributing causality to genetics or medication-induced appetite increases without a clear understanding from healthcare providers.

“*The impact of psychiatric treatments on appetite, leading to unconscious eating, is a significant concern*” (RF5)

Furthermore, the lack of available educational programs exacerbates this knowledge gap, leaving families seeking more information and support.

“*I did not notice this thing; if it exists, they are supposed to deliver information to patients and their families. No one told us there is an education department*” (RM1)

It was suggested that there is a need for holistic care approaches that integrate psychological support into diabetes management.

“*The call for psychological support as part of diabetes care highlights a critical component of holistic patient management*” (RF4)

This thematic analysis underscores the multifaceted barriers encountered in providing comprehensive diabetes care for patients with severe mental illnesses, highlighting areas for systemic improvement and enhanced patient and family education.

#### 3.3.4. Evidence-Based Recommendations for Health Policy Improvement

In addressing the theme of healthcare improvement, participants across the spectrum of medical care provided a wealth of suggestions aimed at elevating the care process for patients grappling with both diabetes and severe mental illnesses.

##### Educational and Awareness Programs

A universal emphasis was placed on the need for robust educational and awareness programs. Stakeholders, including healthcare providers, patients, and their families, were recognized as being integral to the learning process, necessitating targeted courses, lectures, and workshops to enhance the understanding and management of these complex conditions.

Relatives offered perspectives on educational needs, suggesting that more consistent and accessible communication from healthcare providers could significantly enhance care for both patients and their families, as follows:

“*We need doctors to engage in regular communication, providing guidance not just during appointments but also through phone calls, ensuring that we, as caregivers, are well-informed and supported*” (RM3)

##### Specialized Clinics

The call for establishing specialized clinics was underscored by the healthcare providers, who pointed out the need for departments such as nutrition, dermatology, orthopedics, and diabetes. This integration of services is deemed essential in order to improve the quality of care and provide patients with direct and immediate healthcare access.

Also, the desire for an integrated care approach was expressed, with a call for equal emphasis on diabetes care within mental health facilities, as follows:

“*Ensuring that diabetes care is on equal footing with mental health services within the same facility would be a transformative improvement*” (HF1)

##### Advanced Medical Equipment

The participants identified a pressing need for advanced medical equipment, which would not only refine the quality of healthcare services, but also reduce the necessity for external referrals, thereby expediting the treatment process.

##### Facility Environment Improvement

The enhancement of the healthcare environment was a unanimous suggestion, with physicians advocating for improved work settings and the introduction of patient-centered recreational spaces, as follows:

“*For patients facing psychiatric challenges, access to recreational areas like gardens would significantly improve their quality of life and therapeutic experience*” (HF7)

##### Efficient Medical Referral Process

A streamlined referral process was highlighted as a critical area for improvement. The participants agreed that minimizing the time for referral approvals is essential to preventing the deterioration of patients’ health conditions.

##### Staffing

An increase in medical staffing levels, particularly in nursing, was suggested to alleviate current pressures and improve the overall quality of patient care.

##### Patient-Centered Care Perspectives

The patients themselves contributed vital insights, with one emphasizing the significance of compassionate interactions as a means to calm and reassure those in distress, as follows:

“*The usual practice of restraining agitated patients can be traumatic. A more understanding approach that addresses their fears can prevent such situations*” (PM3)

## 4. Discussion

The purpose of this qualitative phenomenological study was to collate experiences of patients with diabetes and severe mental illnesses, their relatives, and healthcare providers to identify barriers to and facilitators of access and use of diabetes care. The themes identified extend the literature on the individual-level, systemic, and socio-cultural barriers to accessing diabetes care among patients with severe mental illnesses.

### 4.1. Identifying Barriers to and Facilitators of Care

The prominence of mental health symptoms in limiting diabetes self-management is consistent with Bellass et al.’s [18] research that outlines the important impact of cognitive and psychological barriers of patients with diabetes and other disorders. However, unlike past studies that largely point to non-adherent behaviors, the present research shows the synergy between stigma and self-esteem factors, consistent with disconnection from care.

Intriguingly, some scholars have postulated that enhanced patient engagement is attainable through targeted psychoeducation among such populations [4,5]. However, the findings from this study identified very low levels of exposure to such services, which indicates a missed opportunity in service provision in these settings. Subsequent studies should assess the reasons for such programs being unavailable and in what ways they might be incorporated more effectively into care trajectories.

Additionally, this study has highlighted several internal barriers, including resource scarcity, the absence of specialized clinics, and staffing shortages. These challenges are consistent with the literature that points to the need for organizational and structural improvements within healthcare systems to better manage diabetes care for those with severe mental illnesses [19,20].

Fragmented care presented itself as a crucial issue, as was evident from the literature on the structural organization of modern healthcare architecture [9]. The findings are most relevant to research conducted in low-resource contexts because patients in such settings frequently lack coordinated healthcare, and this type of disorganization only widens gaps in accessing health care services. On the other hand, models such as the collaborative care model have been reported to be efficient for eradicating such a fragmented approach [12].

As demonstrated from the study area where integrated care is still limited, there are questions for further research on the systemic preparedness, resource commitments, and any measure that defines current healthcare policies. This means that more research needs to be carried out to determine how successful international models can be implemented in the local environment.

Furthermore, the findings showed how cultural stigma affects healthcare engagement. Although Kremers et al.’s [15] study focused on socio-economic status, it explored the challenges faced when implementing universal care and facilitating accessibility to health care services. However, unlike previous studies that mainly explored accessibility due to socio-economic status [13], we described how both caregiver and community stigma can contribute to the process of disengagement further.

One interesting result was that there were gender differences in the cultural norms regarding mental disorders, so it may be necessary for interventions to be culturally tailored. Externally, inadequate familial support and non-adherence among patients emerged as significant barriers. These findings align with research suggesting that the broader social environment and family dynamics play a crucial role in managing chronic conditions alongside mental illness [21].

### 4.2. Recommendations for Service Improvement

The participants advocated developing robust educational programs for medical professionals and patients’ families. This aligns with the literature recognizing education as a cornerstone of effective chronic disease management, predominantly involving coexisting conditions such as severe mental illness [22,23]. For instance, providing mandatory diabetes education for mental health staff will enhance the efficiency of the staff in managing diabetes for patients.

The call for establishing specialized clinics and procuring advanced medical equipment is supported by evidence that suggests that such resources can significantly enhance the quality of care and patient outcomes (HF7). For example, providing specialized diabetes clinics within mental health hospitals will provide more specialized care for patients with severe mental illnesses and diabetes. Similarly, improving the healthcare environment, including creating spaces for patient recreation and relaxation, has been recognized as being beneficial for mental health and overall well-being [24].

Moreover, the need for increased medical staffing, especially in nursing, echoes the sentiment in the literature about the importance of adequate human resources in healthcare delivery (PM3). This is particularly true in mental health settings, where staff–patient ratios can significantly impact the quality of care and patient experience [25].

On the other hand, increased engagement with families and regular follow-up appointments were identified as key areas in need of attention. This recommendation is consistent with research that emphasizes the role of family support in the self-management of chronic illnesses (RM3).

This study’s findings underline the necessity to align with the best practices from physical healthcare settings and to integrate regular training and structured diabetes education within mental health inpatient settings (HF1). This is critical for fostering a healthcare environment conducive to the needs of individuals with dual diagnoses.

Additionally, adopting system- and individual-level actions to enhance medical care for patients with severe mental illnesses is crucial. The literature advocates for psychiatrists to expand their role to include the monitoring and treatment of physical health conditions, such as diabetes, which are prevalent in this patient population [24].

This study corroborates the need for personalized, holistic support that addresses both mental and physical health needs. Strategies involving social support, stress management, better provider communication, and integrated medical and psychiatric care are needed to improve health outcomes for these patients [22,23].

### 4.3. Study Strengths and Limitations

The principal strength of this research is its qualitative methodology, which allowed for a comprehensive exploration of the perspectives and experiences of healthcare providers, relatives, and patients dealing with the dual challenges of diabetes and mental health conditions in Jeddah. This approach yielded rich, nuanced insights that have laid robust groundwork for developing targeted healthcare interventions and strategies.

Nevertheless, the findings’ applicability beyond the local context is tempered by this study’s inherent limitations. The relatively small and context-specific sample size from one region, Jeddah, means that the results may not be readily extrapolated to other settings or populations. Such limitations are characteristic of qualitative research, where a depth of understanding in a specific context is often gained at the expense of broader generalizability. Additionally, interviewing patients with severe mental illnesses has potential biases due to their fluctuating levels of insight or recall.

Nonetheless, the implications of this study are far-reaching, particularly for policy formulation and practice within similar healthcare environments. The depth of insight provided in this study sheds light on the unique challenges faced in managing comorbid diabetes and mental health conditions. These insights offer a valuable template for healthcare policy and practice reform in regions with cultural and healthcare dynamics similar to Jeddah’s. By leveraging the detailed findings of this study, healthcare systems can pilot and evaluate the effectiveness of bespoke interventions tailored to the needs identified through this research. In sum, while this study is contextually bound, its contributions to the field are significant, offering a concrete foundation for potential application and further exploration in related healthcare contexts.

## 5. Conclusions

This study illuminates the intricate challenges and necessary facilitators for effective diabetes management among patients with severe mental illnesses in Jeddah, Saudi Arabia. The findings reveal a critical intersection of healthcare delivery barriers within the system and from external societal factors, underscoring the necessity for a comprehensive, integrated approach to care. Through qualitative insights, this study emphasizes the paramount importance of enhancing educational initiatives for patients and their families, establishing specialized clinics to streamline care, and improving the overall healthcare environment to better accommodate the needs of this vulnerable population. Furthermore, the data points towards the urgent need for systemic reform, including increased staffing levels and the integration of severe mental illness care within diabetes treatment plans, to alleviate the compounded burdens faced by patients. This study’s recommendations highlight the potential for significant advancements in the quality and accessibility of healthcare services, advocating for policies that recognize and address the complex interplay between physical and mental health. Ultimately, the research calls for a holistic model of care that is responsive to the unique challenges at the nexus of diabetes and severe mental illnesses, aiming to foster a healthcare system that is inclusive, effective, and empathetic towards all patients.

## Figures and Tables

**Table 1 healthcare-13-00543-t001:** Participant characteristics for the sample of people with diabetes and SMI.

ID	Primary Diagnosis	Diagnosis Order	Gender	Age	Education Level	Employment Status	**Nationality**
PM1	Schizophrenia	SMI-DM	Male	26	High School	Unemployed	Saudi
PM2	Schizophrenia	SMI-DM	Male	63	High School	Retired	Saudi
PM3	Bipolar disorder	SMI-DM	Male	34	High School	Retired	Saudi
PM4	Bipolar disorder	SMI-DM	Male	46	Fifth grade of primary school	Unemployed	Saudi
PF1	Schizophrenia	SMI-DM	Female	60	Uneducated	Unemployed	Saudi
PF2	Schizophrenia	SMI-DM	Female	70	High School	Unemployed	Saudi
PF3	Schizophrenia	SMI-DM	Female	81	High School	Retired	Saudi
PF4	Schizophrenia	SMI-DM	Female	40	High School	Unemployed	Saudi

Serios Mental Illness (SMI); Diabetes Mellitus (DM).

**Table 2 healthcare-13-00543-t002:** Participant characteristics of the relative’s sample.

ID	Relationship to Patient	Gender	Age	Education Level	Employment Status	Nationality
RM1	Brother	Male	49	High School	Employed	Saudi
RM2	Brother	Male	45	Bachelor’s degree	Employed	Saudi
RM3	Brother	Male	76	Bachelor’s degree	Employed	Saudi
RM4	Brother	Male	36	Bachelor’s degree	Employed	Saudi
RF1	Sister	Female	49	Bachelor’s degree	Employed	Saudi
RF2	Sister	Female	35	Bachelor’s degree	Employed	Saudi
RF3	Sister	Female	38	Bachelor’s degree	Employed	Saudi
RF4	Cousin	Female	43	Bachelor’s degree	Employed	Saudi
RF5	Daughter	Female	22	Bachelor’s degree	Student	Saudi
RF6	Sister	Female	40	Bachelor’s degree	Employed	Saudi

**Table 3 healthcare-13-00543-t003:** Participant characteristics for the healthcare providers’ sample.

ID	Gender	Healthcare Role	Experience Duration
HM1	Male	Diabetologist and endocrinologist	12 years
HM2	Male	Diabetologist and endocrinologist	21 years
HM3	Male	Nurse head	6 years
HM4	Male	Nurse head	4 years
HM5	Male	Nurse	3 years
HF1	Female	Diabetologist and endocrinologist	12 years
HF2	Female	Nurse head	16 years
HF3	Female	Nurse head	13 years
HF4	Female	Nurse	12 years
HF5	Female	Nurse	7 years
HF6	Female	Nurse	10 years
HF7	Female	Nurse	2 years
HF8	Female	Nurse	4 years
HF9	Female	Nurse head	9 years
HF10	Female	Nurse	8 years
HF11	Female	Nurse	13 years
HF12	Female	Nurse	11 years

## Data Availability

Data are publicly available due to the confidentiality of the patients’ information.

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
