# Peer review of "Barriers to Access and Utilization of Diabetes Care Among Patients with Severe Mental Illness in Saudi Arabia: A Qualitative Interpretive Study"

_healthcare, 2025, doi:10.3390/healthcare13050543_

Round 1

Reviewer 1 Report (Previous Reviewer 1)

Comments and Suggestions for Authors

On what basis was the sample size decided?

It would be appropriate to complete the N/A values ​​in Tables 1 and 2.

Which software was used to analyze the post-interview analyses?

What was done in terms of the validity and reliability of the research?

It would be appropriate to visualize the data and add a graph.

Author Response

Comment 1: On what basis was the sample size decided?

Response:

This researcher used saturation data when no new information emerged from the data. As the data collection and analysis were conducted iteratively, the researchers were able to indicted the depth of the information and where the no new themes emerged form the data. So, this approach allowed the researchers to monitor where the researcher should stop.   Please refer to section 2.2, lines from 158-163. 

Comment 2: It would be appropriate to complete the N/A values ​​in Tables 1 and 2.

Response:

We addressed it. Please refer to table 1 and 2

Comment 3: Which software was used to analyze the post-interview analyses?

Response:

The post-interview data was coded manually rather than using software. The rational for analysing the data manually it allows the researchers to engagement  more deeply with the data. 

Comment 4: What was done in terms of the validity and reliability of the research?

Response:

We ensured the validity and reliability of the research through the following:

Validity: We ensured the validity of the study via a in-depth interpretation of findings from the participations. The researchers used iterative approach in collecting the data which allowed engagement with the participations and generate new questions that emerged from the data. Moreover, the researchers communicated to questioned and reflected on the interpretations to ensure better comprehensive.

Reliability: this research use a systematic design to analysis the data via Braun and Clarke  approach. This approach provide a systematic steps to codes the data until discover the enragement themes.

Comment 5: It would be appropriate to visualize the data and add a graph.

Response:

We apologize for not able to provide a visualization of the data, Since we analysis the data manually. 

Reviewer 2 Report (Previous Reviewer 5)

Comments and Suggestions for Authors

Authors improved the manuscript and addressed all my comments. I have no further comments.

Author Response

Comment 1: Authors improved the manuscript and addressed all my comments. I have no further comments.

Response:

Thank you

Reviewer 3 Report (New Reviewer)

Comments and Suggestions for Authors

Peer Review of “Barriers to Access and Utilization of Diabetes Care Among Patients with Mental Illness in Saudi Arabia: A Qualitative Interpretive Study”

Dear Authors,

Below is a detailed peer review of your manuscript. The critique focuses on structural clarity, methodological rigor, coherence between sections, and alignment with your stated research objectives. No additional data collection is suggested; rather, these comments aim to strengthen your existing manuscript.

1. General Comments and Overarching Issues

  1. Consistency in Terminology
    • Your title and some parts of the text refer generally to “Mental Illness,” whereas other sections specify “Severe Mental Illness (SMI).” Clarify and apply one consistent term throughout, particularly in the title and abstract, to avoid confusion about whether all mental illnesses or only severe conditions are included.
  2. Qualitative Approach vs. Phenomenology vs. Interpretive Study
    • The manuscript variously labels the study as a “qualitative phenomenological approach,” an “interpretive study,” and a “thematic analysis” using Braun and Clarke. Phenomenology and thematic analysis are not always synonymous. Strengthen clarity by specifying precisely which methodological tradition you followed and why it was chosen.
  3. Redundancies and Repetitions
    • There are instances of repeated statements on burden/prevalence of diabetes in Saudi Arabia, or repeated references to the significance of mental illness in the Introduction. Consolidate and streamline these to make the manuscript more concise and focused.
  4. Integration of the Conceptual Rationale
    • You state the study is about “barriers to access and utilization of diabetes care,” yet the conceptual or theoretical underpinnings (e.g., stigma theory, integrated care models) are not elaborated. Briefly linking the study’s design and findings to relevant conceptual frameworks would enhance its scholarly depth.

2. Title and Abstract

  • Title: The phrase “Barriers to Access and Utilization…” is appropriate, but ensure it reflects the specific focus on severe mental illness (SMI) if that is truly the target population.
  • Abstract:
    • Aims: The abstract states two aims in overlapping ways: (1) exploring barriers and (2) suggesting evidence-based recommendations. This can be condensed and stated more clearly.
    • Study Design: Specify clearly the type of qualitative approach (e.g., “phenomenological” or “interpretive with thematic analysis”).
    • Results: The results mention four themes but remain very broad. Strengthening how these themes map onto the “barriers” and “recommendations” would highlight your main contributions succinctly.
    • Conclusions: Could better emphasize the study’s unique insight, especially regarding integrated care.

3. Introduction

  1. Clarity of the Gap
    • The Introduction details the prevalence of diabetes and mental illness. While informative, it should more succinctly pinpoint the research gap—for instance, why existing diabetes self-management or education programs are insufficient for individuals with severe mental illnesses in Saudi Arabia.
  2. Literature Synthesis
    • References are current, but the presentation is somewhat repetitive (for instance, you reiterate global statistics and local prevalence multiple times). Consider a more streamlined narrative that:
      • Summarizes global burden and then transitions to Saudi Arabia-specific context.
      • Highlights what we already know about comorbid diabetes and mental illness from prior studies (including elements such as stigma, fragmentation of care).
      • Clarifies what remains unknown or under-explored, particularly in the local Saudi context.
  3. Objective vs. Research Questions
    • You list multiple objectives in the latter part of the Introduction, but they blend barrier identification, facilitator identification, and policy recommendations. Consider presenting them as concise bullet points or structured statements, ensuring each objective aligns directly with your findings in the Results/Discussion.

4. Materials and Methods

  1. Study Design
    • The manuscript uses the term “phenomenological approach,” “qualitative interpretive study,” and then references Braun & Clarke (thematic analysis). These can conflict methodologically. If you are employing thematic analysis in the style of Braun & Clarke, simply label it as “qualitative interpretive approach using thematic analysis,” and remove references to phenomenology unless you explicitly used phenomenological methods (e.g., bracketing, phenomenological reduction).
  2. Sampling and Recruitment
    • You mention that participants (patients with SMI, relatives, and providers) were identified through hospital records, physician referrals, etc. Clarify how many facilities or which type of hospital settings were involved—particularly important for understanding transferability.
    • The paper indicates “data collection continued until no new themes emerged.” Briefly elaborate on how saturation was assessed (e.g., team discussions, concurrent data analysis).
  3. Data Collection
    • You note that 35 interviews (8 patients, 10 relatives, 17 HCPs) yielded about eight hours of total data. For transparency, you might indicate average interview duration per participant group.
    • The interview guide: Provide a short overview of key domains or sample questions (possibly in an appendix). This will help readers assess alignment between your research questions and the data collected.
  4. Data Analysis
    • You reference Braun and Clarke’s six steps, which is good. However, it would strengthen trustworthiness to mention any techniques used (e.g., double coding, peer debriefing, or member checking).
    • Because you analyzed transcripts in Arabic and then translated into English for reporting, indicate how you mitigated potential translation bias (e.g., was there back-translation, or was the research team bilingual?).
  5. Ethical Considerations
    • Ethics approval is clearly stated. It might be helpful to expand on how you handled the mental capacity to consent if participants with severe mental illnesses had episodes of acute psychosis or limited decision-making ability. This clarifies your ethical safeguards.

5. Results

  1. Structure of Themes
    • The themes—(1) Status Quo in Care Provision, (2) Barriers to Access, (3) Navigating Obstacles, and (4) Recommendations—are logical. However, “Navigating the Obstacles” somewhat overlaps with both “Barriers” and “Recommendations.” You might consider merging or clarifying boundaries so each thematic section covers distinct insights.
  2. Participant Characteristics
    • The tables for demographics are helpful but can be tightened. For instance, present only the key demographic factors relevant to the study. Some data columns contain many “N/A” entries, which might be omitted if not applicable.
  3. Illustrative Quotations
    • The quotes are valuable in providing direct participant voices. However, there are occasional repeated references (e.g., “(HF3) says…”). Ensure each subtheme is illustrated with at least one quote from each group (patient, relative, provider) to fully capture the triadic perspective.
  4. Overlap and Repetition
    • Certain sections in “Barriers” repeat statements from “Navigating Obstacles.” For clarity, you could differentiate them by placing all barrier content (individual, systemic, sociocultural) in one section, then all coping or “navigation” strategies in another.

6. Discussion

  1. Depth of Interpretation
    • The Discussion mentions relevant studies but could engage more critically with the novelty or unique aspects of your findings in a Saudi context. For instance, how do stigma and gender norms interplay differently here compared to Western settings?
  2. Alignment with Aims
    • Your stated aims are: (1) identify barriers, (2) identify facilitators, and (3) propose recommendations. The Discussion does address these, but the flow lumps them together in multiple paragraphs. A more structured approach (e.g., “Barriers,” “Facilitators,” “Recommendations for Policy/Practice”) would enhance readability.
  3. Policy Implications and Practice
    • You propose integrated care, specialized clinics, and more staff. Strengthening the “how” of these recommendations (even at a broad level) would give them more weight. For instance, do you suggest a pilot integrated-care unit within existing mental health facilities, or mandatory diabetes education for mental health staff? Concrete suggestions help policymakers and practitioners.
  4. Limitations
    • The limitations are mentioned briefly (context-bound findings, limited sample). Consider clarifying:
      • The transferability question (e.g., single-region data in Jeddah).
      • Potential biases related to interviewing participants with SMI who may have fluctuating levels of insight or recall.

7. Conclusion

  1. Conciseness
    • The Conclusion is thorough but could be made more impactful by summarizing your key findings in a concise manner and reiterating the next steps.
  2. Future Directions
    • You highlight the need for integrated care but might add a short remark on feasible next steps (e.g., implementing a small-scale integrated diabetes–mental health care pilot, or training programs for frontline staff).

8. References and Citations

  1. Reference Formatting
    • Ensure all in-text citations align with the journal’s reference style. Some references appear out of sequence (e.g., duplicates of #18, #19?). Also confirm correct listing order and consistent spacing.
  2. Citation of Qualitative Methodology Sources
    • Given the emphasis on thematic analysis, referencing additional methodological literature (e.g., Braun & Clarke’s updated guidelines or other qualitative reliability/validity sources) could bolster the rigor of the Methods section.

9. Minor Points and Style

  • Language and Grammar: A few sentences in the Introduction and Results have typographical or grammatical slips. A careful final proofread (especially for verb tense consistency) will ensure a polished read.
  • Table Formatting: Some tables stretch across multiple lines, which can disrupt readability. Ensuring each table is cleanly presented with consistent headings will improve presentation.
  •  

Comments on the Quality of English Language
  • Language and Grammar: A few sentences in the Introduction and Results have typographical or grammatical slips. A careful final proofread (especially for verb tense consistency) will ensure a polished read.

Author Response

Comment 1: Consistency in Terminology

    • Your title and some parts of the text refer generally to “Mental Illness,” whereas other sections specify “Severe Mental Illness (SMI).” Clarify and apply one consistent term throughout, particularly in the title and abstract, to avoid confusion about whether all mental illnesses or only severe conditions are included.

Response:

We made the required change to be consistence across the whole research. Please see the change in the title.

Comment 2: Qualitative Approach vs. Phenomenology vs. Interpretive Study

    • The manuscript variously labels the study as a “qualitative phenomenological approach,” an “interpretive study,” and a “thematic analysis” using Braun and Clarke. Phenomenology and thematic analysis are not always synonymous. Strengthen clarity by specifying precisely which methodological tradition you followed and why it was chosen.

Response:

Thank you very much for this formative insights. To clarify, this research conducted qualitative interpretive approach via Braun and Crlack’s thematic analysis  method.

The rational for using Braun and Crlack was to ensure a systematic analysis of the data and discover the emergent themes. We acknowledged that qualitative phenological approach was not accurate in this paper and it will be removed. Please refer to section 2.1, lines from 125- 131, page 3.

Comment 3: Redundancies and Repetitions

    • There are instances of repeated statements on burden/prevalence of diabetes in Saudi Arabia, or repeated references to the significance of mental illness in the Introduction. Consolidate and streamline these to make the manuscript more concise and focused.
    • Repones:
    • The introduction was carefully revised to address the comment. May you look at the revised introduction from 41-115, pages 2&3.

Comment 4: Integration of the Conceptual Rationale

    • You state the study is about “barriers to access and utilization of diabetes care,” yet the conceptual or theoretical underpinnings (e.g., stigma theory, integrated care models) are not elaborated. Briefly linking the study’s design and findings to relevant conceptual frameworks would enhance its scholarly depth.
    • Response:

We appreciate your valuable comment by linking the study deign into conceptual framework. We respect of your comment, we disagree to link the study into conceptual framework like stigma theory or integrated care models. This study is an explored study by using Braun and Clarke thematic analysis approach. So, we are driven by the data to explore the emergent  themes that indicted the barriers and access. Also, we used iterative approach ( as indicted in the manuscript) to reflect and change on the question guide. This approach allowed us to be drive by the data.

  1. Title and Abstract

Comment 5:

Title: The phrase “Barriers to Access and Utilization…” is appropriate, but ensure it reflects the specific focus on severe mental illness (SMI) if that is truly the target population.

  • Response:

Amendment has been made on the title by adding severe mental illness (SMI) within the title, please see the title.

Abstract:

Comment 6: Aims: The abstract states two aims in overlapping ways: (1) exploring barriers and (2) suggesting evidence-based recommendations. This can be condensed and stated more clearly.

    • Response:
    • A change was made. May you look at lines 15-16, in the abstract section, please.
  • Comment 7: Study Design: Specify clearly the type of qualitative approach (e.g., “phenomenological” or “interpretive with thematic analysis”).
    • Response:
    • We addressed the comment and added in the study design more Cleary. Can you refer to the lines 17-20 , please.
  • Comment 8: Results: The results mention four themes but remain very broad. Strengthening how these themes map onto the “barriers” and “recommendations” would highlight your main contributions succinctly.
    • Response:
    • We addressed this comments. We have strengthen our theme to be well aligned with the aim of the study. Can you please refer to the lines 22-25, 346-348, and lines 151,392, 439,and 488.
  • Comment 9: Conclusions: Could better emphasize the study’s unique insight, especially regarding integrated care.

    • Response:
    • Since the word account for the abstract is limited, we have addressed this comment by adding one line regarding the integrated care.  Please refer to the lines 27-28.

  1. Introduction

Comment 10:  Clarity of the Gap

    • The Introduction details the prevalence of diabetes and mental illness. While informative, it should more succinctly pinpoint the research gap—for instance, why existing diabetes self-management or education programs are insufficient for individuals with severe mental illnesses in Saudi Arabia.

    • Response:

See line 84-90.

Comment 11:  Literature Synthesis

    • References are current, but the presentation is somewhat repetitive (for instance, you reiterate global statistics and local prevalence multiple times). Consider a more streamlined narrative that:
      • Summarizes global burden and then transitions to Saudi Arabia-specific context.
      • Highlights what we already know about comorbid diabetes and mental illness from prior studies (including elements such as stigma, fragmentation of care).
      • Clarifies what remains unknown or under-explored, particularly in the local Saudi context.
      • Response:

The introduction was carefully restructured to address the comment. Please refer to the lines 46 to126.

Comment 12: Objective vs. Research Questions

    • You list multiple objectives in the latter part of the Introduction, but they blend barrier identification, facilitator identification, and policy recommendations. Consider presenting them as concise bullet points or structured statements, ensuring each objective aligns directly with your findings in the Results/Discussion.
    • Response:
    • Can you see the revised objectives please. Please refer to the lines 107-112.
  1. Materials and Methods

Comment 13:  Study Design

    • The manuscript uses the term “phenomenological approach,” “qualitative interpretive study,” and then references Braun & Clarke (thematic analysis). These can conflict methodologically. If you are employing thematic analysis in the style of Braun & Clarke, simply label it as “qualitative interpretive approach using thematic analysis,” and remove references to phenomenology unless you explicitly used phenomenological methods (e.g., bracketing, phenomenological reduction).
    • Response:

Thank you very much for this formative insights. To clarify, this research conducted qualitative interpretive approach via Braun and Crlack’s thematic analysis  method.

The rational for using Braun and Crlack was to ensure a systematic analysis of the data and discover the emergent themes. We acknowledged that qualitative phenological approach was not accurate in this paper and it will be removed. Please refer to section 2.1, lines from 125- 131, page 3.

Comment 14:  Sampling and Recruitment

    • You mention that participants (patients with SMI, relatives, and providers) were identified through hospital records, physician referrals, etc. Clarify how many facilities or which type of hospital settings were involved—particularly important for understanding transferability.
    • The paper indicates “data collection continued until no new themes emerged.” Briefly elaborate on how saturation was assessed (e.g., team discussions, concurrent data analysis).
    • Response:
    • 1- There is one hospital in Jeddah city that provides treatment for severe mental illness, however we cannot mention the name in the research for confidential propose.
    • 2- We ensured that the data saturation was achieved through agreement and discussion between the research teams.
  1. Comment 15: Data Collection
    • You note that 35 interviews (8 patients, 10 relatives, 17 HCPs) yielded about eight hours of total data. For transparency, you might indicate average interview duration per participant group.
    • The interview guide: Provide a short overview of key domains or sample questions (possibly in an appendix). This will help readers assess alignment between your research questions and the data collected.
    • Response:
  • Each interview lasted between 40 minutes to one hour. This information was added in the method section, please refer to line 247

  • An appendix was added for topic guide. Please refer to the lines 751 -876

  1. Comment 16: Data Analysis
    • You reference Braun and Clarke’s six steps, which is good. However, it would strengthen trustworthiness to mention any techniques used (e.g., double coding, peer debriefing, or member checking).
    • Because you analyzed transcripts in Arabic and then translated into English for reporting, indicate how you mitigated potential translation bias (e.g., was there back-translation, or was the research team bilingual?).
    • Response:
    • 1- Member checking was conducted to ensure the trustworthiness of the findings. These findings were shared with the research group to give their perspectives and ensure the accuracy of the results. We added in this in the aper. Please refer to the lines 297-299.
    • 2- We ensured the avoidance of translation biases, since all the researchers are bilingual. Moreover,
  1. Comment 17: Ethical Considerations
    • Ethics approval is clearly stated. It might be helpful to expand on how you handled the mental capacity to consent if participants with severe mental illnesses had episodes of acute psychosis or limited decision-making ability. This clarifies your ethical safeguards.
    • Response:
    • Even though the participants with severe mental illnesses might have limited decision-making capacity, their mental capacity were assessed to understand the study’s nature, risks, benefits. However, if they were unable to provide consent, a legally authorized representative or surrogate decision-maker gave consent on their behalf. Additionally, the participant’s assent were sought if possible, and their well-being were continuously monitored throughout the study. See line 274-279.
  1. Results

Comment 18:  Structure of Themes

    • The themes—(1) Status Quo in Care Provision, (2) Barriers to Access, (3) Navigating Obstacles, and (4) Recommendations—are logical. However, “Navigating the Obstacles” somewhat overlaps with both “Barriers” and “Recommendations.” You might consider merging or clarifying boundaries so each thematic section covers distinct insights.
    • Response:
    • We belief that the title of themes do not reflect of the title. So, we considered to rename the themes to be  well aligned with the title of the paper. Please refer to the lines 22-25, 346-348, and lines 151,392, 439,and 488.

Comment 19: Participant Characteristics

    • The tables for demographics are helpful but can be tightened. For instance, present only the key demographic factors relevant to the study. Some data columns contain many “N/A” entries, which might be omitted if not applicable.
    • Response:
    • Completed, please refer to table 1 and 2.

Comment 20: Illustrative Quotations

    • The quotes are valuable in providing direct participant voices. However, there are occasional repeated references (e.g., “(HF3) says…”). Ensure each subtheme is illustrated with at least one quote from each group (patient, relative, provider) to fully capture the triadic perspective.

Response:

Thank you very much for your valuable comments. We disagree for the following reason:

As the natural of the qualitative study and driven by the data, the quotes were included equally across the themes to ensure better representation of the sample. Although, some sub-themes have not quotations from particular sample, the results provide comprehensive representation of all sample of the study.

Comment 21: Overlap and Repetition

    • Certain sections in “Barriers” repeat statements from “Navigating Obstacles.” For clarity, you could differentiate them by placing all barrier content (individual, systemic, sociocultural) in one section, then all coping or “navigation” strategies in another.
    • Response:
    • As we understand, we have already differentiate between the two section. If you refer to theme 3.3.2 &3.3.3, the two sections were separated.

  1. Discussion

Comment 22:  Depth of Interpretation

    • The Discussion mentions relevant studies but could engage more critically with the novelty or unique aspects of your findings in a Saudi context. For instance, how do stigma and gender norms interplay differently here compared to Western settings?
    • Response:
    • Could you please refer to the lines where you want us to modify?

Comment 23:  Alignment with Aims

    • Your stated aims are: (1) identify barriers, (2) identify facilitators, and (3) propose recommendations. The Discussion does address these, but the flow lumps them together in multiple paragraphs. A more structured approach (e.g., “Barriers,” “Facilitators,” “Recommendations for Policy/Practice”) would enhance readability.
    • Response:
    • In the discussion section we started with the Identifying Barriers and facilitators to Care together in one section then followed by the Recommendations for Service Improvement section for more clarity.
    •  
  1. Comment 24: Policy Implications and Practice
    • You propose integrated care, specialized clinics, and more staff. Strengthening the “how” of these recommendations (even at a broad level) would give them more weight. For instance, do you suggest a pilot integrated-care unit within existing mental health facilities, or mandatory diabetes education for mental health staff? Concrete suggestions help policymakers and practitioners.
    • Response:

See lines 618-620, 623-625,

  1. Comment 25: Limitations
    • The limitations are mentioned briefly (context-bound findings, limited sample). Consider clarifying:
      • The transferability question (e.g., single-region data in Jeddah).
      • Potential biases related to interviewing participants with SMI who may have fluctuating levels of insight or recall.
      • Responses:

May you refer to lines 652-657.

  1. Conclusion

Comment 26: Conciseness

    • The Conclusion is thorough but could be made more impactful by summarizing your key findings in a concise manner and reiterating the next steps.
  1. Comment 27: Future Directions
    • You highlight the need for integrated care but might add a short remark on feasible next steps (e.g., implementing a small-scale integrated diabetes–mental health care pilot, or training programs for frontline staff).
    • Response:
    • Please refer to lines 618-620, 623-625.

  1. References and Citations

Comment 28:  Reference Formatting

    • Ensure all in-text citations align with the journal’s reference style. Some references appear out of sequence (e.g., duplicates of #18, #19?). Also confirm correct listing order and consistent spacing.
  1. Comment 29: Citation of Qualitative Methodology Sources
    • Given the emphasis on thematic analysis, referencing additional methodological literature (e.g., Braun & Clarke’s updated guidelines or other qualitative reliability/validity sources) could bolster the rigor of the Methods section.
    • References:

As requested to reference the most updated Braun & Clarke’s guidelines. We substitute the old guideline (2006) by the most recent publication guideline (2012) to enhance the method section. Please refer to reference 18.

  1. Minor Points and Style
  • Comment 30: Language and Grammar: A few sentences in the Introduction and Results have typographical or grammatical slips. A careful final proofread (especially for verb tense consistency) will ensure a polished read.
  • Response:

We tried our best to check the language.

  • Comment 31:Table Formatting: Some tables stretch across multiple lines, which can disrupt readability. Ensuring each table is cleanly presented with consistent headings will improve presentation.
  •  Response:
  • Please refer to the table and we hope that we arranged them as requested.

Comments on the Quality of English Language

  • Comment 32: Language and Grammar: A few sentences in the Introduction and Results have typographical or grammatical slips. A careful final proofread (especially for verb tense consistency) will ensure a polished read.
  • Response:
  • We tried our best to check the language.

Round 2

Reviewer 3 Report (New Reviewer)

Comments and Suggestions for Authors

I have no further comments. After taking a second loom, the authors have addressed and revised most of the issues raised in my early comment

The paper can be accepted as it is with minor English correction which can be done during copy editing

Author Response

Comment 1: The paper can be accepted as it is with minor English correction which can be done during copy editing 

Response: 

Thank you very much for taking the time to revise your manuscript and provide your valuable insights into our manuscript.

This manuscript is a resubmission of an earlier submission. The following is a list of the peer review reports and author responses from that submission.

Round 1

Reviewer 1 Report

Comments and Suggestions for Authors

One of the best examples of qualitative research. However, it would be useful to explain some points.

Which computer-aided software was used for coding and analysis?

(Nvivo, Atlas.ti, MaxQDA, QDAMiner and HyperResearch)

The analysis methods and techniques applied in these types of research should be given in detail. Which of the following was used?

Phenomenological Analysis?

Content Analysis?

Descriptive Analysis?

Grounded Theory and Constant Comparison Analysis?

Discourse Analysis?

Ethnomethodology?

Author Response

Comment 1: One of the best examples of qualitative research. However, it would be useful to explain some points.

Response:

Thank you very much.

Comment 2: Which computer-aided software was used for coding and analysis?

(Nvivo, Atlas.ti, MaxQDA, QDAMiner and HyperResearch)

Response:

The data was coded manually rather than using software. The rational for analysing the data manually it allows the researchers to engagement  more deeply with the data. 

Comment 3: The analysis methods and techniques applied in these types of research should be given in detail. Which of the following was used?

Phenomenological Analysis?

Content Analysis?

Descriptive Analysis?

Grounded Theory and Constant Comparison Analysis?

Discourse Analysis?

Ethnomethodology?

Response:

A thematic analysis (Braun and Clarke 6 steps) was used to analysis the content. Please refer to line 219.

Reviewer 2 Report

Comments and Suggestions for Authors

An article on a topical and very interesting subject that hasn't been explored much. Pleasant to read and methodologically correct.

Objective title, although a little long.

Summary - simple, I think the theoretical basis could be a little more developed.

Keywords - Some are not included in the Mesh descriptors. Suggestion: introduce Type 1 Diabetes and replace Healthcare Barriers with Barriers to Access.

Introduction - good grounding and contextualisation of the topic and clear identification of the study's objectives.

From a methodological point of view, they refer to the qualitative approach and the content analysis method used, and in point 2.5 they identify the four stages of Braun and Clarke's method.

Well-defined inclusion criteria for users, relatives and health professionals.

They make reference to the ethical issues that have been met and the absence of conflicts of interest. 

They clearly state the study sample, presenting the sociodemographic characterisation of all those involved.

In line 140 they put iterative but I think this is a mistake.

They identify the 4 areas/themes that emerged from the content analysis and develop them by complementing them with extracts from the interviews.

They could have addressed the guiding question(s) of the interview and the average length of the interviews.

In the discussion, they try to explore the emerging themes using the theoretical framework.

Identify the limitations of the study and its impact, emphasising the advantage of using qualitative approaches in research.

Objective and well-written conclusion.

Very up-to-date bibliographical references, with 18 dated 2020 or more.

Author Response

Comment 1: An article on a topical and very interesting subject that hasn't been explored much. Pleasant to read and methodologically correct.

Reponses:

 thank you

Comment 2: Objective title, although a little long.

Response:

Firstly, we agree that the title of the article is long. As a result, the title was shortens. Please refer to the lines 2-4 in editing document.

Secondly, the objective was also shortness as requested, please refer to the line  

Comment 3: Summary - simple, I think the theoretical basis could be a little more developed.

Response: The study used a qualitative phenomenological approach.

See lines: 148-153.

Keywords - Some are not included in the Mesh descriptors. Suggestion: introduce Type 1 Diabetes and replace Healthcare Barriers with Barriers to Access.

Response: The change was made as requested. Please refer to the line 26 in editing document.

Comment 4: Introduction - good grounding and contextualisation of the topic and clear identification of the study's objectives.

Response: Thank you.

Comment 5: From a methodological point of view, they refer to the qualitative approach and the content analysis method used, and in point 2.5 they identify the four stages of Braun and Clarke's method.

Response:

If you refer to line 138, we mentioned that aggregate of codes (which is mostly content analysis). According to the Braun and Clarke.

Comment 6: Well-defined inclusion criteria for users, relatives and health professionals.

Response: thank you

Comment 7: They make reference to the ethical issues that have been met and the absence of conflicts of interest. 

Response: Thank you

Comment 8: They clearly state the study sample, presenting the sociodemographic characterisation of all those involved.

Response:

Thank you

Comment 9: In line 140 they put iterative but I think this is a mistake.

Response:

Disagree, we belief that iterative approach is important as allow for depth understanding of the data. However, iterative approach allowed to reflect on the data and made changes on interview guides.

Comment 10: They identify the 4 areas/themes that emerged from the content analysis and develop them by complementing them with extracts from the interviews.

Response:

Yes, the four themes were emerged from the content analysis. Also, we supported the themes by the direct quotes from the participants.

Comment 11: They could have addressed the guiding question(s) of the interview and the average length of the interviews.

Response:

The topics guide was added in the appendix.  

Comment 12: In the discussion, they try to explore the emerging themes using the theoretical framework.

Response:

This section was entirely rewritten as requested by the other authors.

Comment 13: Identify the limitations of the study and its impact, emphasising the advantage of using qualitative approaches in research.

Response:

Thank you.

Comment 14: Objective and well-written conclusion.

Response:

Thank you

Comment 15: Very up-to-date bibliographical references, with 18 dated 2020 or more.

Response:

Thank you

Reviewer 3 Report

Comments and Suggestions for Authors

Thank you for the opportunity to review this manuscript.  The following comments and questions are submitted with all due respect to the authors.

-Line 39:  The authors reference, “significant variability in diabetes prevalence across different age groups and localities.”  It would be beneficial to support this statement with data.

-Line 45: The authors may wish to spell out SMI the first time this abbreviation is used.  Also, it would likely help the reader to provide an operational definition of SMI.  In addition, the authors occasionally use “SMI” and other times refer to “mental health conditions” and other terms referring to “mental health.”  The authors may wish to review this to increase consistence and clarity.

-Paragraph (Lines 46-55):   The authors may wish to review this paragraph and review for clarity.

-Lines 66-72:  The authors wish to identify factors that could improve care, which would be beneficial.  Do the authors also wish to identify barriers?

-Line 92:  Type 1 diabetes is mentioned, but the focus on this manuscript is on type 2 diabetes.  Would suggest maintaining this focus (on type 2 diabetes) and specifying for the author that this is the focus because, for example, it is the most prevalent type of diabetes.

-Line 94:  How many were completed in each language?  In the methods, the authors note that the information was initially reviewed in Arabic.

-Line 98:  The authors may wish to note why “financial support” is an important consideration.

-Line 123:  It appears that the authors are referring to a semi-structured interview when stating “pre-established topic guide.”  It may help to simple state that they utilized a semi-structured interview.

-Lines 138-139:  Was ‘translation, back-translation’ utilized?  How did the authors “[ensure] the translation preserved the original meaning and context”?

-Line 157-158:  For some participants, N/A is indicate for age.  Was this information not available in the medical chart?

-Table 1:  Authors may wish to include ‘duration since diagnosis” as an average time since diagnosis is given in Lines 184-185.

-Table 1:  If the focus is on Type 2 diabetes and this is clear in the methods, the column for “Diabetes Type” may be deleted.

-Table 2:  Do the authors believe that it would be helpful to state how long the relatives have been caring for their person with diabetes?

-Table 3:  Review last entry under "Experience duration” as there appears to be a typo.

-Lines 176, 196, 261:  Authors may wish to follow the professional guidelines around the use of diabetes language and avoid referring to people with diabetes as “diabetic."

-Line 185:  For clarify, were all 35 individuals with diabetes in this study “insulin dependent”?

-Line 199:  The authors may wish to clarify (by offering examples) what is meant by “necessary investigations”).

-LINE 205:  Are “dietary regulations” separate from the “treatment plan” or a part of it?

-Line 214:  This section focuses on nurses.  When “home care team” is mentioned, is this referring to a team of nurses only?

-Line 220:  Was the statement about “leave for lectures” made by a nurse?  Are lectures referring to patient education for the 35 people with diabetes included in this study?

-The authors may wish to review and edit the Results section to increase clarity.

-Line 236:  The authors may wish to specify that some medications (vs. “these medications”) may very well associated with increase risk for diabetes.

-Lines 241-245:  Unclear if this statement is referring to psychotropic medication(s).

-Lines 252-255:  The authors may wish to add references to support statements.

-Line 254:  The authors may wish to refer to blood glucose (vs. “blood sugar”) for consistence.

-Lines 256-259:  The authors may wish to add references to support statements.  In addition, the authors introduce additional variables including mobility and sexual desire.  If these are major factors in this study, the authors may wish to include relevant information and references in the introduction.

-Lines 281-282:  The authors may wish to add references to support this statement. 

-Lines 281-302:  The authors may wish to review and edit this section to increase clarity.

-Line 378:  The authors may wish to use a term such as “suboptimal” vs. “non-adherence."

-Lines 431, 438-440:  The authors are urged to consider using more tentative language because stating that “the implications of this study are far-reaching” and that this study “[offers] a concrete foundation...” does not reflect the exploratory nature of this qualitative study based on a small sample size.

Comments on the Quality of English Language

Minor editing of English language required.  For example, ghe authors may wish to edit to increase clarity:  Lines 37, 65, 85, 90, 122, 150, Table 2 heading.

Author Response

Comment 1: Comment 1: Thank you for the opportunity to review this manuscript.  The following comments and questions are submitted with all due respect to the authors.

Response:

Thank you for your valuable feedback which we believe that will improve the quality of the paper.

Comment 2: Comment 2: -Line 39:  The authors reference, “significant variability in diabetes prevalence across different age groups and localities.”  It would be beneficial to support this statement with data.

Repones:

A supportive data was added to indicate the variability in diabetes prevalence across different age and localities in Saudi Arabia. Please refer to the lines between 55-59.

Comment 3: Comments-Line 45: The authors may wish to spell out SMI the first time this abbreviation is used.  Also, it would likely help the reader to provide an operational definition of SMI.  In addition, the authors occasionally use “SMI” and other times refer to “mental health conditions” and other terms referring to “mental health.”  The authors may wish to review this to increase consistence and clarity.

Response:

We had already spelled out SMI at the first paragraph of the introduction, please refer to line 43. Also, the full words were added in the abstract for better clarity to the reader. A clear definition of serious mental illness was added as recommended . Please refer to the lines between 51-54.

Comment 4: the authors occasionally use “SMI” and other times refer to “mental health conditions” and other terms referring to “mental health.”  The authors may wish to review this to increase consistence and clarity.

Response:

To increase consistency and clarity we used term SMI in the whole paper.

Comment 5: Comments: -Paragraph (Lines 46-55):   The authors may wish to review this paragraph and review for clarity.

Response: This paragraph was reviewed carefully for better clarity. Please refer to lines 71-82 in the edited manuscript.

Comment 6: Comments -Lines 66-72:  The authors wish to identify factors that could improve care, which would be beneficial.  Do the authors also wish to identify barriers?

Response:

One paragraph has been added to barriers and facilitators  of accessing care among patients with types 2 diabetes with SMI. Please refer to lines 132-140.

Comment 7: -Line 92:  Type 1 diabetes is mentioned, but the focus on this manuscript is on type 2 diabetes.  Would suggest maintaining this focus (on type 2 diabetes) and specifying for the author that this is the focus because, for example, it is the most prevalent type of diabetes.

Response:

Change has been made and emphasis on type 2 diabetes as suggested.

Comment 8: -Line 94:  How many were completed in each language?  In the methods, the authors note that the information was initially reviewed in Arabic.

Response:

All interviews were conducted in Arabic language since all the participants are speaking Arabic language only. Yes, it was clarified in the data analysis section lines: 228-231.

Comment 9: -Line 98:  The authors may wish to note why “financial support” is an important consideration.

Response:

The reason was added in line 173.

Comment 10: -Line 123:  It appears that the authors are referring to a semi-structured interview when stating “pre-established topic guide.”  It may help to simple state that they utilized a semi-structured interview.

Response:

The sentence was fixed in line 212.

Comment 11: -Lines 138-139:  Was ‘translation, back-translation’ utilized?  How did the authors “[ensure] the translation preserved the original meaning and context”?

Response:

The authors decided to translate the interviews as literally as possible, making only minor adjustments to correct grammar and enhance the clarity of the English translation as all authors are flaunt in English. Moreover, the transcripts were revised by two authors to ensure the clarity of the transcript.

Comment 12: -Line 157-158:  For some participants, N/A is indicate for age.  Was this information not available in the medical chart?

Response:

The information is not available in the medical file.

Comment 13: -Table 1:  Authors may wish to include ‘duration since diagnosis” as an average time since diagnosis is given in Lines 184-185.

Response:

It’s included. See line 292.

Comment 14: -Table 1:  If the focus is on Type 2 diabetes and this is clear in the methods, the column for “Diabetes Type” may be deleted.

Response:

The column for diabetes type was deleted. Please refer to table 1.

Comment 15: -Table 2:  Do the authors believe that it would be helpful to state how long the relatives have been caring for their person with diabetes?

Response:

We believe this information is not important since it does not add value to the results

Comment 16: -Table 3:  Review last entry under "Experience duration” as there appears to be a typo.

Response:

We edited the typo.

Comment 17: -Lines 176, 196, 261:  Authors may wish to follow the professional guidelines around the use of diabetes language and avoid referring to people with diabetes as “diabetic."

Response:

As we must follow the professional guideline, diabetic words were changed. Please refer to the edited document in the  lines 281-282, 309-310 and 419.

Comment 18: -Line 185:  For clarify, were all 35 individuals with diabetes in this study “insulin dependent”?

Response:

Yes. See lines 291-292.

Comment 19:-Line 199:  The authors may wish to clarify (by offering examples) what is meant by “necessary investigations”).

Response:

It was clarified see lines 312-314.

Comment 20: -LINE 205:  Are “dietary regulations” separate from the “treatment plan” or a part of it?

Response:

It was fixed. See lines 317-319.

Comment 21: -Line 214:  This section focuses on nurses.  When “home care team” is mentioned, is this referring to a team of nurses only?

Response:

Home care team refer to the nurses. See line 328.

Comment 22: -Line 220:  Was the statement about “leave for lectures” made by a nurse?  Are lectures referring to patient education for the 35 people with diabetes included in this study?

Response:

Refers to nurses’ education. See line 334.

Comment 23: -The authors may wish to review and edit the Results section to increase clarity.

Response:

The section was reviewed and rewritten.

Comment 24: -Line 236:  The authors may wish to specify that some medications (vs. “these medications”) may very well associated with increase risk for diabetes.

Response:

Section was modified entirely. See lines 388-394.

Comment 25: -Lines 241-245:  Unclear if this statement is referring to psychotropic medication(s).

Response:

This statement refer to the impact of psychotropic medications on the patients such as gaining weight and this is stated clearly.

Comment 26: -Lines 252-255:  The authors may wish to add references to support statements.

Response:

Thank you for your valuable insights and appreciate your guidance. We disagree to integrate a references from the literature, since the result represent the insights from the participants and not discuss how these findings relate to the existing literatures. The references to support statements are added in the discussion section.

Comment 27: -Line 254:  The authors may wish to refer to blood glucose (vs. “blood sugar”) for consistence.

Response:

This section was deleted and rewritten entirely. See line 377.

Comment 28: -Lines 256-259:  The authors may wish to add references to support statements.  In addition, the authors introduce additional variables including mobility and sexual desire.  If these are major factors in this study, the authors may wish to include relevant information and references in the introduction.

Response:

Thank you for your valuable insights and appreciate your guidance. We disagree to integrate a references from the literature, since the result section represent the insights from the participants and not discuss how these findings relate to the existing literatures. However, This section was modified and rewritten.

Comment 29: -Lines 281-282:  The authors may wish to add references to support this statement. 

Response:

Thank you for your valuable insights and appreciate your guidance. We disagree to integrate a references from the literature, since the result section represent the insights from the participants and not discuss how these findings relate to the existing literatures. You will find this in the discussion section.

Comment 30: -Lines 281-302:  The authors may wish to review and edit this section to increase clarity.

Response:

This section was edited , please refer to the lines from 443-486, please in the edited manuscript.

Comment 31: -Line 378:  The authors may wish to use a term such as “suboptimal” vs. “non-adherence."

Response:

The section was rewritten entirely.

Comment 32: -Lines 431, 438-440:  The authors are urged to consider using more tentative language because stating that “the implications of this study are far-reaching” and that this study “[offers] a concrete foundation...” does not reflect the exploratory nature of this qualitative study based on a small sample size.

Response:

We would argued that, first, the sample of this study is small, however, this study was conducted inductively allowing for deeper understanding of the topic. Moreover, this study conducted an iterative approach which allow for deeper understanding of the participants perspectives.

Reviewer 4 Report

Comments and Suggestions for Authors

Barriers to Access and Utilization of Diabetes Care Among Patients with Mental Illness: A Qualitative Interpretive Study on Patients, Relatives, and Healthcare Providers' Perspectives

Materials and Methods

2.2. Participant Selection and Criteria

- Please provide further details regarding the sources from which the three groups of participants were chosen.

- Please provide further details regarding the conditions or criterias that participants selected for mental health illness are required to have?

2.4. Data Collection Procedures

- Has triangulation been conducted to gather data? Kindly provide additional clarification.

Result

- The title is "Barriers to Access and Utilization of Diabetes Care"; however, the results inadequately address the barriers to access and utilization of diabetes care, despite interviews conducted with patients, relatives, and medical personnel, which should provide comprehensive insights into barriers from various viewpoints. Recommendations for enhancing this section

- In the case where the theme has a lot of details. The details of each theme may be divided into subthemes to facilitate understanding for the reader.

Discussion: The written discussion must be rewritten in their entirety.

- The discussion should be rewritten to correspond with the study's findings and the character of the writing discussion in the manuscript. By addressing significant issues, including comparisons with other studies and the rationale for their similarities or differences with other research.

- Study Strengths and Limitations should be move to the discussion section.

Author Response

Reviewer 4

Materials and Methods

2.2. Participant Selection and Criteria

Comment 1: - Please provide further details regarding the sources from which the three groups of participants were chosen.

Response:

Details were added. Please see lines 166-177.   

Comment 2: - Please provide further details regarding the conditions or criterias that participants selected for mental health illness are required to have?

Response:

Details were added. Please see lines 165-168.    

2.4. Data Collection Procedures

Comment 3:- Has triangulation been conducted to gather data? Kindly provide additional clarification.

Response:

Clarification was provided. See lines 203-2011.

Result

Comment 4:- The title is "Barriers to Access and Utilization of Diabetes Care"; however, the results inadequately address the barriers to access and utilization of diabetes care, despite interviews conducted with patients, relatives, and medical personnel, which should provide comprehensive insights into barriers from various viewpoints. Recommendations for enhancing this section

Response:

Section was modified and an entire section was added related to barriers to access and utilization of diabetes care among patients with severe mental illness. Please see lines 378- 421.

Comment 5: - In the case where the theme has a lot of details. The details of each theme may be divided into subthemes to facilitate understanding for the reader.

Response:

The whole section was rewritten.

Discussion: The written discussion must be rewritten in their entirety.

Comment 6: - The discussion should be rewritten to correspond with the study's findings and the character of the writing discussion in the manuscript. By addressing significant issues, including comparisons with other studies and the rationale for their similarities or differences with other research.

Response:

The section was rewritten entirely.

Comment 7: - Study Strengths and Limitations should be move to the discussion section.

Response:

The section was rewritten entirely and the Study Strengths and Limitations moved into the discussion section as requested. Please refer to line 689 in the edited manuscript.

Reviewer 5 Report

Comments and Suggestions for Authors

1.     Title Improvement: The title should be revised to include the country name.

2.     Abstract Clarity: A. The abstract lacks clear information about the perspectives of patients, relatives, and healthcare providers, despite the title mentioning these groups. Ensure the abstract elaborates on how the study captures these perspectives. B. It would be helpful to include details on when and where the study was conducted.

3.     Introduction Justification: The rationale for studying the perspectives of patients, relatives, and healthcare providers is unclear. Please elaborate on why understanding these perspectives is essential for improving mental health care and overcoming barriers to treatment.

4.     Methods Section: A. The sampling method, which included 35 participants to reach theoretical saturation, raises concerns about whether this is sufficient to represent the three different groups. If the sample started primarily with patients, it seems unlikely that saturation could be easily achieved. Please clarify and justify the sample size for each group. B. If semi-structured interviews were used, provide more details on how they were initiated. For instance, did interviews start by discussing about diabetes first, or did they focus on mental illness initially?

5.     Results Section: In the results, the general characteristics are placed under the subtitle "Thematic Analysis and Emerging Themes" (3.2). It might be better to separate the general characteristics from the thematic analysis to maintain a clearer structure.

6.     Discussion Section: The discussion relies heavily on one study that aligns with the authors' analysis. It would be more insightful to explore studies that do not align with these findings and explain potential discrepancies. The discussion should also be rewritten to form more cohesive paragraphs, providing a thorough review of related literature.

7.     Minor: A. In the Line 13, before the abbreviation "SMI," please add the word "severe" so that it reads as "severe SMI." B. In Table 3, correct "ears" to "years."

Author Response

Comment 1: Title Improvement: The title should be revised to include the country name.

Response:

The title was revised and included the Country’s name. Please refer to line 3 in the edited manuscript.

Comment 2:   2.     Abstract Clarity: 

  1. The abstract lacks clear information about the perspectives of patients, relatives, and healthcare providers, despite the title mentioning these groups. Ensure the abstract elaborates on how the study captures these perspectives. 
  2. Itwould be helpful to include details on when and where the study was conducted.

Response:

  • In the method section, we added information about the participants, Please refer to the lines 20-23 in the edited manuscript. We ensured to elaborate how the study captured the perspectives via including details of data analysis process. Please refer to the lines 21-23 in the edited manuscript.
  • The date and place were added in lines 22-23.

Comment 4:     3.     Introduction Justification: The rationale for studying the perspectives of patients, relatives, and healthcare providers is unclear. Please elaborate on why understanding these perspectives is essential for improving mental health care and overcoming barriers to treatment.

Response:

Justification was added. Please see lines 96-107, 137-145.

Comment 4:    4.     Methods Section: 

  1. The sampling method, which included 35 participants to reach theoretical saturation, raises concerns about whether this is sufficient to represent the three different groups. If the sample started primarily with patients, it seems unlikely that saturation could be easily achieved. Please clarify and justify the sample size for each group.

  1. If semi-structured interviews were used, provide more details on how they were initiated. For instance, did interviews start by discussing about diabetesfirst, or did they focus on mental illness initially?

Response:

  • Sample size justification was provided. Please see lines 210-216.
  • Interviews questions were added in the appendix.

Comment 5: 5.     Results Section: In the results, the general characteristics are placed under the subtitle "Thematic Analysis and Emerging Themes" (3.2). It might be better to separate the general characteristics from the thematic analysis to maintain a clearer structure.

Response:

           A different section was added related to the general characteristics of the participants. Please refer     to line 304 in the edited manuscript.  Moreover, under the Thematic Analysis and Emerging Themes, we listed the emerging themes for better clarity to the readers.

Comment 6: 6.     Discussion Section: The discussion relies heavily on one study that aligns with the authors' analysis. It would be more insightful to explore studies that do not align with these findings and explain potential discrepancies. The discussion should also be rewritten to form more cohesive paragraphs, providing a thorough review of related literature.

Response:

The section was rewritten entirely.Please refer to the

Comment 7:   7.     Minor: A. In the Line 13, before the abbreviation "SMI," please add the word "severe" so that it reads as "severe SMI." B. In Table 3, correct "ears" to "years."

Response:

  • We added in the word severe. Please refer to line 14 in the edited manuscript.
  • We corrected the word from ears to years.